# A SARS-CoV-2 variant elicits an antibody response with a shifted immunodominance hierarchy

**Allison J. Greaney**[1,2]*, **Tyler N. Starr**[1,3], **Rachel T. Eguia**[1], **Andrea N. Loes**[1,3], **Khadija Khan**[4,5], **Farina Karim**[4,5], **Sandile Cele**[4,5], **John E. Bowen**[6], **Jennifer K. Logue**[7], **Davide Corti**[8], **David Veesler**[3,6], **Helen Y. Chu**[7], **Alex Sigal**[4,5], **Jesse D. Bloom**[1,3]*

**1** Basic Sciences Division and Computational Biology Program, Fred Hutchinson Cancer Research Center, Seattle, Washington, United States of America, **2** Department of Genome Sciences & Medical Scientist Training Program, University of Washington, Seattle, Washington, United States of America, **3** Howard Hughes Medical Institute, Chevy Chase, Maryland, United States of America, **4** Africa Health Research Institute, Durban, South Africa, **5** School of Laboratory Medicine and Medical Sciences, University of KwaZulu–Natal, Durban, South Africa, **6** Department of Biochemistry, University of Washington, Seattle, Washington, United States of America, **7** Division of Allergy and Infectious Diseases, University of Washington, Seattle, Washington, United States of America, **8** Humabs BioMed SA, a subsidiary of Vir Biotechnology, Bellinzona, Switzerland

* agreaney@fredhutch.org (AJG); bloom@fredhutch.org (JDB)

**Data Availability Statement:** The complete code for the full computational data analysis pipeline of the mapping experiments is available at https://github.com/jbloomlab/SARS-CoV-2-RBD_B.1.351.

## Abstract

Many SARS-CoV-2 variants have mutations at key sites targeted by antibodies. However, it is unknown if antibodies elicited by infection with these variants target the same or different regions of the viral spike as antibodies elicited by earlier viral isolates. Here we compare the specificities of polyclonal antibodies produced by humans infected with early 2020 isolates versus the B.1.351 variant of concern (also known as Beta or 20H/501Y.V2), which contains mutations in multiple key spike epitopes. The serum neutralizing activity of antibodies elicited by infection with both early 2020 viruses and B.1.351 is heavily focused on the spike receptor-binding domain (RBD). However, within the RBD, B.1.351-elicited antibodies are more focused on the "class 3" epitope spanning sites 443 to 452, and neutralization by these antibodies is notably less affected by mutations at residue 484. Our results show that SARS-CoV-2 variants can elicit polyclonal antibodies with different immunodominance hierarchies.

## Author summary

SARS-CoV-2 has circulated among humans for approximately two years, and mutations in emerging variants can erode immunity elicited by prior infection or vaccination. Our understanding of the antibody response elicited by these new variants is still limited. For other viruses, such as influenza, antigenically drifted variants can elicit antibodies that target different sites. Here, we find that this principle also applies to SARS-CoV-2. While the "class 2" RBD antibody epitope is immunodominant for sera from donors infected with

The escape fraction measured for each mutation in S3 Data and also at https://github.com/jbloomlab/SARS-CoV-2-RBD_B.1.351/blob/main/results/supp_data/B1351_raw_data.csv. All raw sequencing data are available on the NCBI Short Read Archive at BioProject PRJNA770094, BioSample SAMN22208699, SAMN22208700. The neutralization titers of vaccine- and infection-elicited sera against the tested RBD point mutants is at https://github.com/jbloomlab/SARS-CoV-2-RBD_B.1.351/blob/main/experimental_data/results/neut_titers/neut_titers.csv.

**Funding:** This project has been funded in part with federal funds from the NIAID/NIH (https://www.niaid.nih.gov/) under contract numbers HHSN272201400006C and 75N93021C00015, to J.D.B. This work was supported by grants from the NIAID / NIH (R01AI141707 to J.D.B., and T32AI083203 to A.J.G., DP1AI158186 and HHSN272201700059C to D.V.), a Pew Biomedical Scholars Award (https://www.pewtrusts.org/) (D. V.), an Investigators in the Pathogenesis of Infectious Disease Awards from the Burroughs Wellcome Fund (https://www.bwfund.org/) (D.V.), and the Gates Foundation (https://www.gatesfoundation.org) (INV-004949 to J.D.B, INV-016575 to H.Y.C. OPP1156262 to D.V., and INV-018944 to A.S.). The Scientific Computing Infrastructure at Fred Hutch is funded by ORIP (https://orip.nih.gov/) grant S10OD028685. T.N.S. is a Howard Hughes Medical Institute Fellow of the Damon Runyon Cancer Research Foundation (https://www.damonrunyon.org/) (DRG-2381-19). J.D.B. is an Investigator of the Howard Hughes Medical Institute (https://www.hhmi.org/). H.Y.C. is also funded by an Emergent Ventures Award (https://www.mercatus.org/emergent-ventures). The funders had no role in study design, data collection and analysis, decision to publish, or preparation of the manuscript.

**Competing interests:** I have read the journal's policy and the authors of this manuscript have the following competing interests: J.D.B. consults for Moderna and Flagship Labs 77 on topics related to viral evolution, and has the potential to receive a share of IP revenue as an inventor on a Fred Hutch optioned technology/patent (application WO2020006494) related to deep mutational scanning of viral proteins. A.J.G., T.N.S., and J.D.B have the potential to receive a share of IP revenue as an inventor on a Fred Hutch optioned technology related to deep mutational scanning of the receptor-binding domain of SARS-CoV-2 Spike protein. H.Y.C. is a consultant for Merck, Pfizer, Ellume, and the Bill and Melinda Gates Foundation and has received support from Cepheid and Sanofi-

SARS-CoV-2 in early 2020, antibodies elicited by infection with the B.1.351 (Beta) variant are more focused on the "class 3" epitope. Notably, the class 3 epitope is conserved between the early 2020 and B.1.351 viruses, but is mutated in the Delta variant, which rose to high frequency globally in mid-2021. As SARS-CoV-2 continues to circulate among humans, individuals' prior infection and vaccination histories may partially determine their susceptibilities to viral mutants in new variants.

## Introduction

Over the past year, SARS-CoV-2 viral variants have emerged with mutations that alter the antigenicity of spike and erode neutralization of the virus by infection- and vaccine-elicited polyclonal antibodies [1–13]. While it is well established that many SARS-CoV-2 variants are less susceptible to antibody immunity generated by early 2020 infections, it is unknown if the antibodies elicited by infection with these variants have different specificities and epitope immunodominance hierarchies. For influenza virus, it has been demonstrated that immunodominance of different epitopes changes over time as the virus evolves antigenically [14–16]. If a similar phenomenon occurs for SARS-CoV-2, then the sites of important antigenic mutations will change over time.

Here we address this question by combining serology and deep mutational scanning to compare the specificity of the polyclonal antibody response elicited by infection with early 2020 viruses versus the B.1.351 variant (also referred to as Beta or 20H/501Y.V2). The B.1.351 variant was first detected in Nelson Mandela Bay, South Africa and likely emerged in August 2020 after the country's first epidemic wave [17]. B.1.351 was the dominant lineage in South Africa by the end of 2020, although it has subsequently been displaced by the B.1.617.2 (Delta) lineage [18]. B.1.351 has mutations throughout the spike protein, including at key epitopes in both the RBD and NTD [1,2,6,9]. The B.1.351 variant has among the largest reductions in neutralization by convalescent plasmas of any SARS-CoV-2 variant to date [7,8,19–21]. Additionally, prior work has demonstrated that B.1.351 convalescent plasmas can neutralize early 2020 viruses better than early 2020 plasmas can neutralize B.1.351 viruses [1,22], suggesting that there may be a shift in the specificity of the antibody response. Our results described below expand this understanding by showing that while neutralization by B.1.351-elicited plasma antibodies is still heavily focused on the RBD, their site-specificity within the RBD is somewhat shifted compared to antibodies elicited by early 2020 viruses. Specifically, within the RBD, B.1.351-elicited sera is relatively more targeted to the class 3 epitope (in the classification scheme of [23]) and relatively less targeted to the class 1 and 2 epitopes.

## Results

### The B.1.351 SARS-CoV-2 variant lineage has mutations in multiple spike epitopes

The B.1.351 spike used in our experiments contained the following mutations relative to the Wuhan-Hu-1 strain: D80A, D215G, del242–244, K417N, E484K, N501Y, D614G, and A701V (**Fig 1**); note that some B.1.351 viruses also contain L18F. Three of these mutations are in the RBD (K417N, E484K, and N501Y). K417N and E484K strongly disrupt binding of class 1 and class 2 antibodies, respectively [24]. N501Y is in or proximal to the class 3 epitope, but does not strongly affect the binding or neutralization of polyclonal convalescent or vaccine-elicited

Pasteur. D.V. is named as an inventor on a patent application filed by the University of Washington related to SARS-CoV-2 vaccines and has received an unrelated sponsored research agreement from Vir Biotechnology Inc. D.C. is an employee of and may hold shares in Vir Biotechnology. The other authors declare no competing interests.

antibodies [8,23], although it enhances the RBD's affinity for its receptor, angiotensin converting enzyme 2 (ACE2) [11,25,26].

## Convalescent plasma samples from individuals infected with B.1.351 or an early 2020 virus

We obtained plasma samples collected approximately 30 days post-symptom onset (mean 33, range 27–40 days) from 9 individuals infected with SARS-CoV-2 during the "second wave" of COVID-19 in South Africa from late December 2020 through late January 2021 (Table 1). During this timeframe, B.1.351 virus accounted for >90% of sequenced infections in the area [1,17,18]. None of the individuals had evidence of prior SARS-CoV-2 infection, so we presume these individuals experienced a primary B.1.351 infection.

To enable comparison of B.1.351-elicited antibodies to those elicited by infection with an early 2020 virus, we reexamined a set of convalescent plasma samples collected approximately 30 days post-symptom onset (mean 32, range 15–61 days) from 17 individuals with symptom onset on or prior to March 15, 2020 in Washington State, USA (Table 1) [27,28]. At that time, most sequenced viral isolates in Washington State had spike sequences identical to Wuhan-Hu-1, although D614G viruses were also present at a low level [29,30]. No other spike mutations were present at appreciable frequencies at that time.

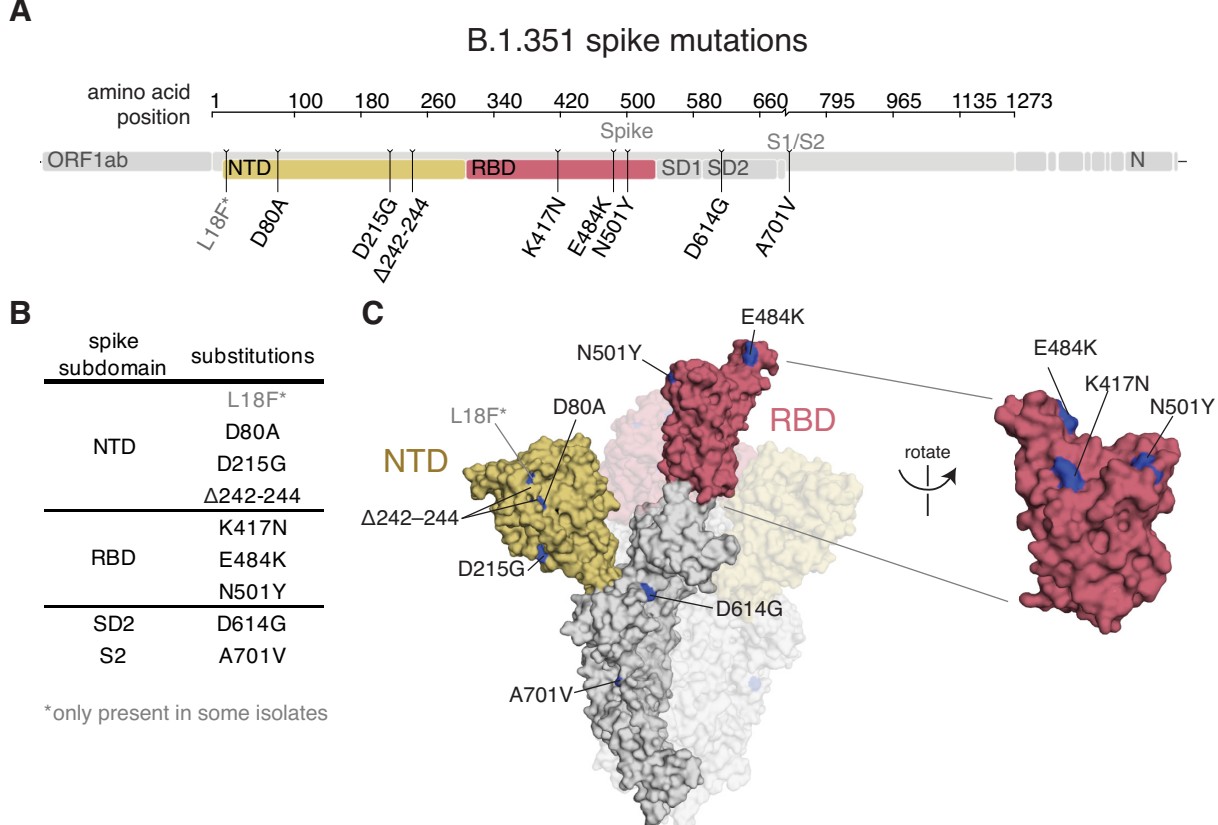

**Fig 1. B.1.351 spike mutations. (A, B)** Mutations in the B.1.351 spike relative to Wuhan-Hu-1 [17]. L18F is only present in some B.1.351 isolates. Visualization generated by https://covdb.stanford.edu/sierra/sars2/by-patterns/. **(C)** Sites where mutations occur in the spike ectodomain are highlighted in blue on the Wuhan-Hu-1 one-RBD open spike trimer (left, PDB 6ZGG) [75] or RBD (right) (PDB 6M0J) [76]. The surface of one spike monomer is shown; the other two protomers are transparent.

**Table 1. Information on cohorts of individuals infected with early 2020 or B.1.351 viruses.**

| Infecting Virus | Time Period | Location | Days Post-Symptom Onset | Number of Individuals |
|---|---|---|---|---|
| Early 2020 | Prior to March 15, 2020 | Washington State, USA | mean 32 (range 15–61) | 17 |
| B.1.351 | Late December 2020 to late January 2021 | South Africa | mean 33 (range 27–40) | 9 |

## Infection with B.1.351 elicits a neutralizing antibody response at least as RBD-focused as early 2020 viruses

Early 2020 viruses induce a neutralizing antibody response that largely targets the RBD [28,31,32], although some neutralizing antibodies also bind the NTD [9,33–35]. Because B.1.351 has mutations in both the RBD and NTD, it is important to determine if the specificity of the neutralizing antibody response elicited by this virus is similarly RBD-focused.

We depleted plasmas from B.1.351-infected individuals of B.1.351 RBD-binding antibodies, or performed a mock depletion, and measured neutralization of B.1.351 spike-pseudotyped lentiviral particles (**Figs 2A** and **S1** and **S1 Data**). The median neutralization titer (NT50) of these plasmas against the B.1.351-spike-pseudotyped lentiviral particles for the mock depletion was 2,459 (range 259–5,081). For 7 out of 9 samples, greater than 90% of neutralizing activity was ablated by removal of RBD-binding antibodies (**Fig 2A**).

We compared these B.1.351 results to previous measurements of the RBD-focused neutralizing activity of plasmas from individuals infected with early 2020 viruses. These prior measurements were made using Wuhan-Hu-1 RBD depletions and D614G spike-pseudotyped lentiviral particles [28]. The neutralizing activity of the B.1.351 plasmas was at least as RBD-focused as the early 2020 virus plasmas, with most neutralizing activity of most plasmas from both cohorts attributable to RBD-binding antibodies (**Fig 2B and 2C**). There was a slight trend for the neutralizing activity of the B.1.351 plasmas to be more RBD-focused than the early 2020 plasmas, but the difference was not statistically significant (**Fig 2C**). One caveat is that all neutralization assays were performed in 293T cells overexpressing ACE2, which tend to emphasize the effect of RBD-binding, ACE2-competitive antibodies more than assays performed on cells with lower levels of ACE2 expression [7,35,36].

## Complete mapping of mutations in the B.1.351 RBD that reduce binding by polyclonal plasma antibodies elicited by B.1.351 infection

To determine how mutations within the RBD affect plasma antibody binding, we used a previously described deep mutational scanning approach. Briefly, this approach involves generating comprehensive mutant libraries of the RBD, displaying the mutant RBDs on the surface of yeast, and using fluorescence-activated cell sorting (FACS) and deep sequencing to quantify how mutations impact antibody binding [28,37].

Previously, we have performed such deep mutational scanning using the RBD from the Wuhan-Hu-1 isolate to map mutations that affect binding by polyclonal antibodies elicited by infection or vaccination that involves a RBD identical to that in Wuhan-Hu-1 [24,28,38]. However, for the current work we wanted to determine the specificity of antibodies elicited by B.1.351 infection to the B.1.351 RBD. Therefore, we generated new duplicate libraries containing 99.7% (3,807 of 3,819) of the possible single amino-acid mutations in the B.1.351 RBD. We displayed these libraries on the surface of yeast, and measured the effects of mutations on RBD expression and binding to ACE2 (**S2 Fig** and **S2 Data** [25]). We used computational filters based on these measurements as well as a pre-sort of the library for RBDs that bind ACE2 with at least 1% the avidity of the unmutated B.1.351 RBD to filter spurious antibody-escape mutations that were highly deleterious or led to gross unfolding of the RBD.

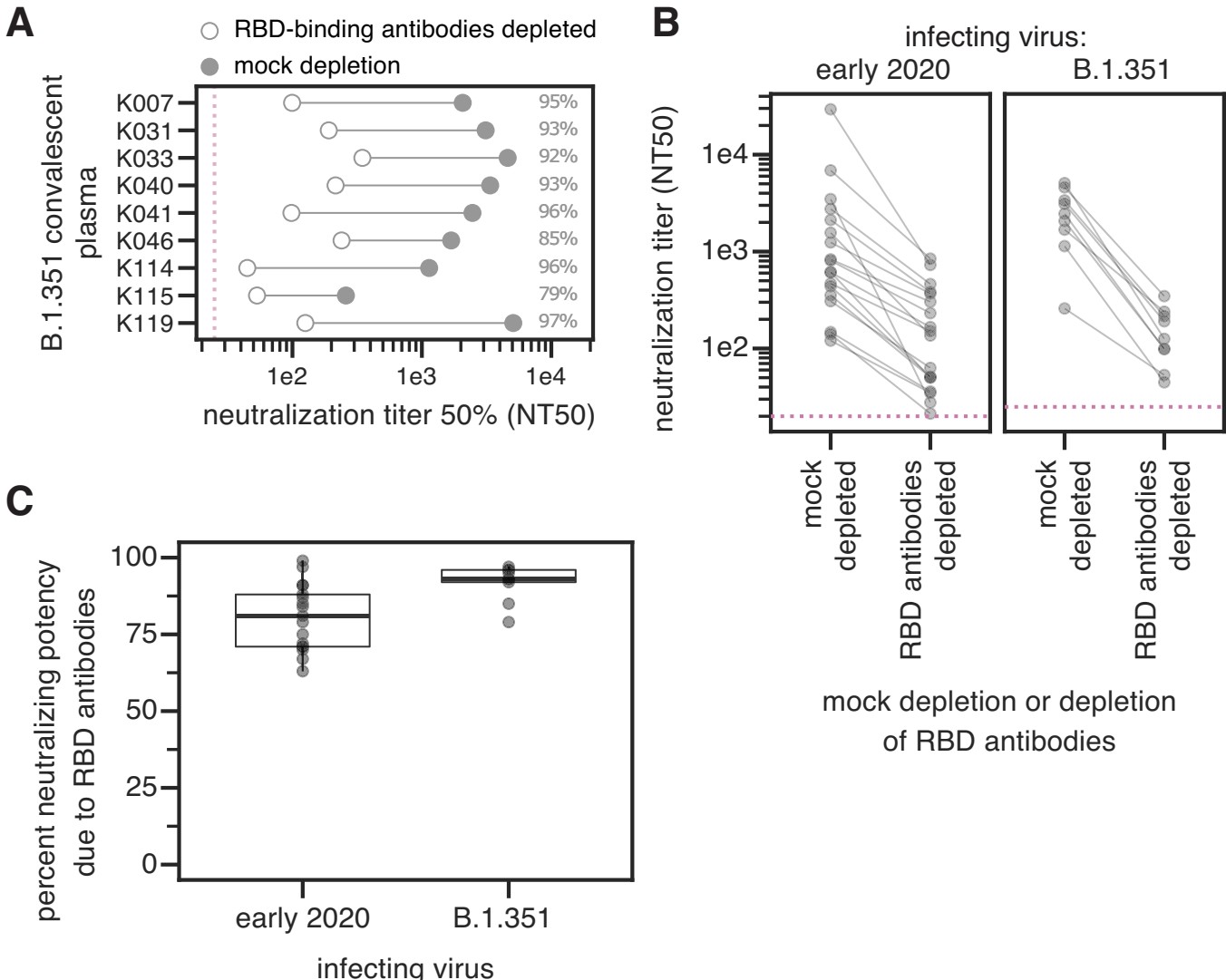

**Fig 2. The neutralizing activity of plasma antibodies elicited by B.1.351 infection is heavily focused on the RBD. (A)** The neutralizing titer (NT50) of plasmas from B.1.351-infected individuals against B.1.351 spike-pseudotyped lentiviral particles, following mock depletion or depletion of B.1.351 RBD-binding antibodies. **(B)** Comparison of neutralization titer following mock depletion or depletion of B.1.351 RBD-binding antibodies for early 2020 (n = 17) [28] and B.1.351 convalescent plasmas (n = 9). The pink dashed line in A, B indicates the limit of detection (NT50 of 25 for B.1.351 plasmas, and 20 for early 2020 plasmas). **(C)** Percent loss of neutralization after removal of RBD-binding antibodies for early 2020 and B.1.351 convalescent plasmas. The difference is not significant (Cox proportional-hazards test, accounting for censoring, p = 0.12). Experiments with B.1.351 infection-elicited plasmas were performed with B.1.351 RBD proteins and spike-pseudotyped lentiviruses, and experiments with early 2020 plasmas were performed with Wuhan-Hu-1 RBD proteins and D614G spike-pseudotyped lentiviruses. The data for the early 2020 viruses are reprinted from [28]. Neutralization titers are in **S1 Data** and at https://github.com/jbloomlab/SARS-CoV-2-RBD_B.1.351/blob/main/experimental_data/results/rbd_depletion_neuts/RBD_depletion_NT50_b1351_haarvi.csv. Full neutralization curves for the B.1.351 plasmas are in **S1 Fig**, and the full curves for the early 2020 plasmas are shown in the supplement of [28].

We then measured how all the single RBD mutations affected the binding of polyclonal antibodies in the B.1.351 convalescent plasmas to the B.1.351 RBD. To do this, we incubated the yeast-displayed B.1.351 libraries with each plasma and used fluorescence-activated cell sorting (FACS) to enrich for RBD mutants with reduced antibody binding as measured using an IgG+IgA+IgM secondary antibody (**S3A–S3C Fig**). FACS selection gates are set to capture the approximately 5% of cells with the lowest amount of antibody binding for their amount of RBD expression. This involves some subjectivity, which may affect which mutations are

identified as antibody-escape variants. We deep sequenced the pre- and post-enrichment populations to quantify each mutation's "escape fraction". These escape fractions range from 0 (no cells with the mutation in the escape bin) to 1 (all cells with the mutation in the plasma-escape bin) (**S3 Data**). The escape fractions measured for independent biological replicate libraries were well-correlated (**S3D Fig**), and in the sections below we report the average across the two replicate libraries. We represent the escape maps as logo plots, where the height of each letter is proportional to its escape fraction (**Figs 3 and S3A**).

## B.1.351-elicited antibodies focus on different epitopes than early 2020 convalescent samples

We examined the sites and epitopes to which mutations had the greatest effect on antibody binding. We use the Barnes, et al. [23] antibody epitope classification scheme, in which there are antibody classes 1 through 4 (**Fig 3A**). The class 1, 2, and 3 antibodies are often potently neutralizing, while the class 4 antibodies are usually less potently neutralizing *in vitro* [31–33,39,40]. Relative to Wuhan-Hu-1, B.1.351 contains mutations in or proximal to the class 1, 2, and 3 epitopes (K417N, E484K, and N501Y, respectively) (**Fig 3A**), although the N501Y mutation has little effect on polyclonal convalescent antibody binding or neutralization for Wuhan-Hu-1-like viruses [7,8,41].

For the B.1.351 plasmas, in 4 of 9 cases, mutations to site 484 within the class 2 epitope had the largest effects on antibody binding and the K484E reversion mutation had little effect (**Fig 3B and S3 Data**). In 3 of 9 cases, mutations to the class 3 epitope (sites 443–450, 498–501, shown in cyan) and the class 2 site 484 had comparably large effects on antibody binding. In two cases, no mutation had a particularly large effect on binding. Mutations to the class 1 and 4 antibody epitopes did not have large effects on plasma binding.

There are clear differences in the RBD epitope targeting of the B.1.351 plasmas versus previously characterized plasmas from a cohort of individuals (n = 11) infected with early 2020 viruses in Washington State, USA [28]. These 11 samples are a subset of the 17 whose RBD-targeting neutralizing activity is described above, chosen to cover a range of serum binding and neutralizing potencies and degrees of RBD-directed neutralization potencies (**Fig 2B and 2C**) [28]. Specifically, binding of the early 2020 plasmas were most affected by mutations to the class 1 and 2 epitopes, with mutations to sites 456, 486, and 484 having some of the largest effects on binding to the RBD (**Figs 4 and S4 and S4 Data**), although mutation to site 456 have little effect on neutralization *in vitro* reflecting the common hyperfocusing of neutralizing antibody responses [28,38]. While the B.1.351 plasmas were also strongly affected by mutations to the class 2 epitope and site 484, mutations to the class 1 epitope had little effect. Moreover, while both groups of plasmas are affected by class 3 epitope mutations, the relative importance of class 3 mutations is greater for the B.1.351 plasmas (**Fig 4A and 4B**).

There is also heterogeneity among the antibody-escape maps within each of the two cohorts as well as similarities between cohorts. For instance, the antibody-escape map for participant C of the early 2020 cohort qualitatively resembles that of the "484-focused" B.1.351 cohort samples, and the maps for participants G and H qualitatively resemble the "484 and class 3-focused" group. Thus, the trends observed here must be interpreted with the caveat that the two cohort sizes are relatively small.

## Class 3 epitope mutations have a larger effect on neutralization for B.1.351 plasmas, while mutations at the class 2 site 484 have a larger effect for early 2020 plasmas

To test if the differences in plasma antibody binding specificity described above lead to different effects of mutations on neutralization, we performed neutralization assays on key mutants

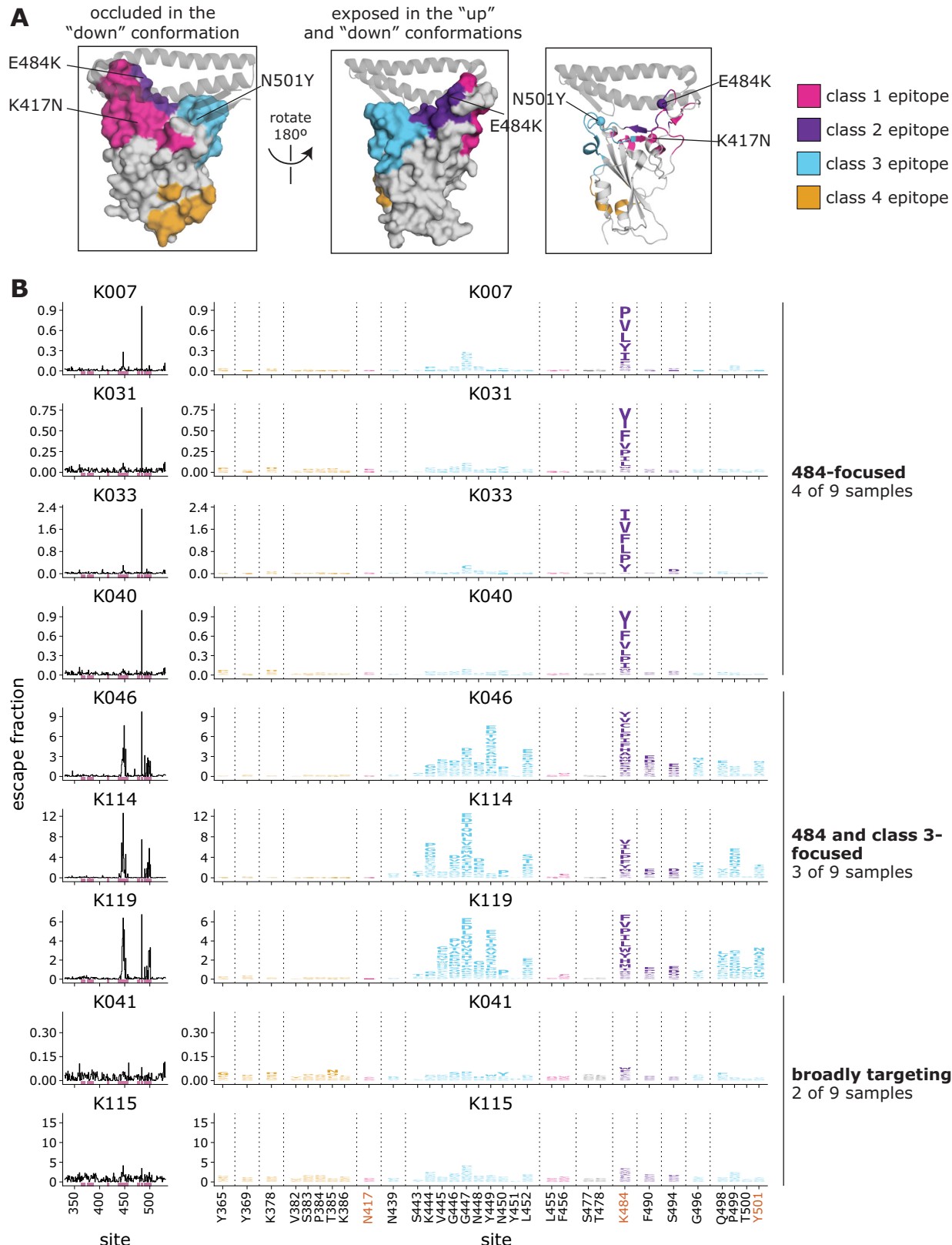

**Fig 3. Complete maps of mutations in the B.1.351 RBD that reduce binding by B.1.351 convalescent plasmas. (A)** The Wuhan-Hu-1 RBD (PDB 6M0J) colored by antibody epitope [23]. The three sites where mutations distinguish the Wuhan-Hu-1 and B.1.351 RBDs are labeled. ACE2

is shown as a gray ribbon diagram. **(B)** Escape maps for B.1.351 convalescent plasmas. The line plots at left indicate the sum of effects of all mutations at each RBD site on plasma antibody binding, with larger values indicating more escape. The logo plots at right show key sites (highlighted in purple on the line plot x-axes). The height of each letter is that mutation's escape fraction; larger letters indicate a greater reduction in binding. For each sample, the y-axis is scaled independently. RBD sites are colored by epitope as in (A). Sites 417, 484, and 501 are labeled with red text on the x-axis. All escape scores are in **S3 Data** and at https://github.com/jbloomlab/SARS-CoV-2-RBD_B.1.351/blob/main/results/supp_data/B1351_raw_data.csv. Interactive versions of logo plots and structural visualizations are at https://jbloomlab.github.io/SARS-CoV-2-RBD_B.1.351/.

using spike-pseudotyped lentiviral particles. For these experiments, we chose the eight B.1.351 samples with the highest neutralizing potency (there was not enough residual sample volume to perform neutralization assays with the lowest-potency sample). We also chose four early 2020 samples with substantial RBD-focused neutralizing activity and with antibody-binding escape maps representative of the early 2020 cohort as a whole (**S4 Fig**). In all assays, we tested neutralization by B.1.351 and early 2020 plasmas against point mutants in the homologous B.1.351 or D614G spikes.

Mutations to site 484 had strikingly different effects on neutralization by B.1.351 versus early 2020 plasmas. For the early 2020 plasmas, both E484K/Q mutations, as well as the K417N-E484K-N501Y triple mutation, reduced neutralization by >10-fold, which is comparable to the reduction caused by removing all RBD-binding antibodies from the plasmas (**Fig 5**). Therefore, the neutralizing activity of early 2020 plasmas is often highly focused on site 484, as has been described previously [2,7,8,28,38,41–44]. In contrast, mutations to site 484 had much smaller effects on neutralization by B.1.351 plasmas. The K484E reversion had little effect on neutralization by B.1.351 plasmas, which was striking given the large effect of E484K on early 2020 plasma neutralization. While the K484Q mutation had the largest effect on B.1.351 plasmas of any of the single mutations we tested (geometric mean of 3.0-fold change), the effect was smaller than that for the early 2020 plasmas (geometric mean of 18.3-fold change).

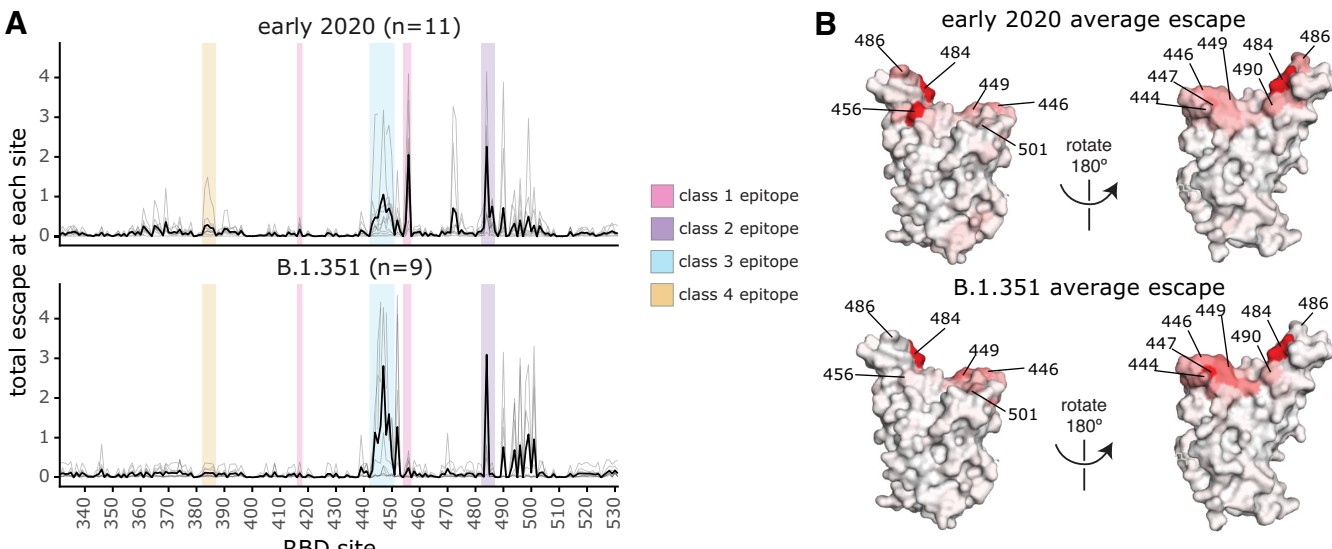

**Fig 4. Comparison of binding escape mutations between plasmas elicited by infection with B.1.351 versus early 2020 viruses. (A)** The total escape at each site is shown as a light gray line for each plasma in the early 2020 or B.1.351 cohorts. The thicker black line indicates the average for each cohort. Key antibody epitopes are highlighted, colored as in **Fig 2A**. **(B)** The total escape at each site averaged across each cohort is mapped to the Wuhan-Hu-1 RBD surface (PDB 6M0J [76]), with sites colored from white to red, with white indicating no escape, and red being the site with the most escape. Interactive versions of logo plots and structural visualizations are at https://jbloomlab.github.io/SARS-CoV-2-RBD_B.1.351/. The early 2020 escape-mapping data in this figure were originally published in [28] and are reanalyzed here. The full escape maps for the early 2020 samples are shown in **S4 Fig** and the full escape maps for the B.1.351 samples are shown in **Fig 3**.

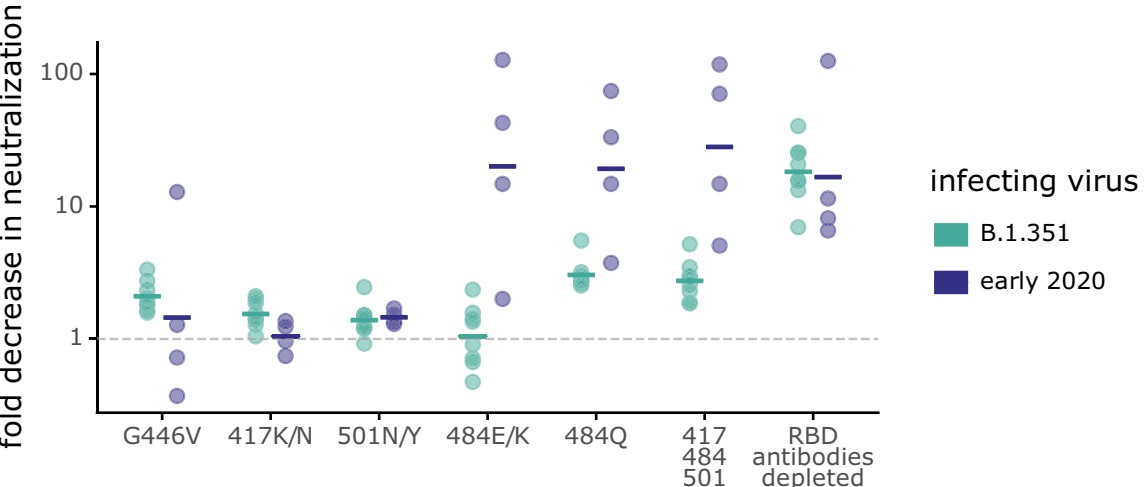

**Fig 5. Some mutations have different effects on neutralization by B.1.351 and early 2020 plasmas.** Plasmas from B.1.351- or early 2020-convalescent individuals were tested for neutralization of wildtype B.1.351 or D614G spike-pseudotyped lentiviral particles, respectively, and against the indicated point mutants in their respective parental backgrounds. The y-axis indicates the fold-change in neutralization caused by the mutations, with larger values indicating less neutralization. Each point is the average of two technical replicates for one individual. The crossbars indicate the group geometric mean. The dashed gray line is at 1 (i.e., mutation causes no change in neutralization). Sites 417, 484, and 501 differ between B.1.351 and early 2020 viruses, and so mutations are tested in each background that changes the identity to that in the other virus (e.g., E484K in early 2020 viruses, and K484E in B.1.351). Full neutralization curves and effects of mutations for each individual are shown in **S5 Fig**, and the numerical values and IC50s are given in **S5 Data** and at https://github.com/jbloomlab/SARS-CoV-2-RBD_B.1.351/blob/main/experimental_data/results/neut_titers/neut_titers.csv.

The class 3 epitope was a slightly more important target of neutralization for the B.1.351 plasmas than for early 2020 plasmas, consistent with the deep mutational scanning escape maps. The G446V mutation to the class 3 epitope had a slightly larger, but still modest, effect on neutralization for the B.1.351 plasmas than for most of the early 2020 plasmas (**Figs 5 and S5**). No tested mutation, nor the 417-484-501 triple mutant, reduced neutralization by the B.1.351 plasmas as much as removing all RBD-binding antibodies (**Fig 5**), a result in stark contrast to that observed for the early 2020 plasmas.

## Discussion

We found that a SARS-CoV-2 variant induces antibody responses with different immunodominance hierarchies than early SARS-CoV-2 viral isolates. Changes in immunodominance hierarchies over time and asymmetric antigenic drift have also been observed for influenza virus [14–16,45]. Such changes can have important consequences, as they can contribute to individuals with different exposure histories having different susceptibilities to viral mutants [46,47]. Although the changes in immunodominance we have observed here are relatively modest, they could become larger as the virus continues to evolve and different individuals accumulate increasingly disparate exposure histories through infection and vaccination [48,49].

We suggest several speculative hypotheses about several reasons why B.1.351 might elicit different hierarchies of antibodies. Although the B.1.351 spike protein has multiple mutations in key antigenic sites in the RBD and NTD [1,2], the neutralizing antibody response elicited by B.1.351 infection is at least as RBD-focused as for early 2020 infections, suggesting that none of the RBD mutations have reduced the antigenicity of that spike subdomain. But within the RBD, site 484 may be less immunodominant for B.1.351-elicited plasmas. Specifically, the E484K and E484Q mutations, which have large effects on early 2020 plasmas, have more moderate effects on

neutralization by B.1.351-elicited plasmas. Infection with early 2020 viruses frequently leads to the development of neutralizing class 2 antibodies that target an epitope containing site 484 [24,42,50,51], and are derived from common antibody germline genes (e.g., IGHV3-53/66, IGHV3-30, IGHV1-2 [23,52,53]. We speculate that viruses containing K484 rather than E484 (such as B.1.351) might less readily elicit such neutralizing antibodies [54], or might elicit antibodies that draw less of their binding energy from site 484. Furthermore, if the class 2 epitope (containing site 484) is less immunogenic in B.1.351, that could lead to relatively stronger targeting of the class 3 epitope for B.1.351-elicited sera. Note that such phenomena could be human-specific, since the class 2 epitope containing site 484 is not as immunodominant in other species with different germline antibody genes (i.e., rhesus macaques) [55].

Changing immunodominance hierarchies could explain previous reports that polyclonal antibodies elicited by infection with different SARS-CoV-2 variants can have differing neutralizing breadths and specificities [4,56–58]. For instance, prior studies of individuals infected with B.1.351 demonstrated that the convalescent plasmas from B.1.351-infected individuals neutralized early 2020 viruses better than early 2020 convalescent plasmas neutralized B.1.351 viruses [1,22]. Our results help mechanistically explain this finding by showing that one of the key epitopes that differs between early 2020 viruses and B.1.351 (the class 2 epitope centered on site 484) is more immunodominant for early 2020 infections. Such changes in immunodominance hierarchies could also explain recent results suggesting that polyclonal antibodies elicited by B.1.351 infection are less effective at neutralizing the Delta (B.1.617.2) variant than antibodies elicited by early 2020 viruses [4,59].

Our study has several limitations. The cohorts of individuals infected with early 2020 and B.1.351 viruses are small, and are geographically and temporally distinct. Specifically, the early 2020 samples were collected in early 2020 in Washington State, USA, and the B.1.351 samples were collected in December 2020–January 2021 in South Africa. Nevertheless, the two cohorts are relatively well-matched with respect to age, sex, and days-post symptom onset of sample collection (**Table 1**) and assays were performed under comparable conditions. But host factors, including antibody germline gene alleles, immune history, and prior exposures to endemic coronaviruses may contribute to the differences observed in the specificity of the SAR-CoV-2 antibody response. Our deep mutational scanning measured binding to yeast-displayed RBD, which may not capture all relevant features of full-length spike in the context of virus. Finally, our neutralization assays used pseudotyped lentiviral particles and ACE2-overexpressing cells, and some recent works suggest that the relative importance of different spike epitopes for neutralization can depend on the viral system and target cell line used [7,35,36,60].

Although the B.1.351 variant has now been displaced, our results illustrate the need to understand immunity elicited by different SARS-CoV-2 variants. As population immunity due to infection or vaccination increases, preexisting immunity is becoming an increasingly important driver of SARS-CoV-2 evolution [61], as has shown to be the case for seasonal coronaviruses [62,63]. Moreover, as individuals begin to accumulate more complex SARS-CoV-2 immune histories due to multiple infections and/or vaccinations, the effects of immune imprinting or original antigenic sin [64,65] may start to interact with the variant-specific immunodominance hierarchies we have described to create increasingly diverse antibody specificities in the human population.

## Methods

### Ethics statement

Samples were collected from participants enrolled in a prospective cohort study approved by the Biomedical Research Ethics Committee (BREC) at the University of KwaZulu–Natal

(reference BREC/00001275/2020) or a prospective longitudinal cohort study in Seattle, WA, approved by the University of Washington Institutional Review Board (protocol #STUDY00000959). Written informed consent was obtained from each participant.

## Materials availability

The SARS-CoV-2 RBD mutant libraries and unmutated parental plasmid are available upon request with completion of an MTA. The plasmid encoding the SARS-CoV-2 spike gene used to generate pseudotyped lentiviral particles, HDM_Spikedelta21_D614G, is available from Addgene (#158762) and BEI Resources (NR-53765). The HDM_Spikedelta21_B.1.351 plasmid is available upon request. Further information and requests for reagents and resources should be directed to and will be fulfilled by Jesse Bloom (jbloom@fredhutch.org) upon completion of a materials transfer agreement.

## Description of cohort

Samples were collected from participants enrolled in a prospective cohort study approved by the Biomedical Research Ethics Committee (BREC) at the University of KwaZulu–Natal (reference BREC/00001275/2020). Written informed consent was obtained from each participant. The mean age was 54 years (median 53; range 26–78 years). Four were males and 5 were females. All participants had symptomatic SARS-CoV-2 infection and a positive SARS-CoV-2 qPCR from a swab of the upper respiratory tract, and all participants required hospitalization. All 9 participants were HIV-negative. None of the participants had evidence of prior SARS-CoV-2 infection. Blood was sampled approximately 30 days post-symptom onset (mean 32.9, range 27–40 days) from 9 individuals infected with SARS-CoV-2 during the "second wave" of infections in South Africa from late December 2020 through late January 2021, when the B.1.351 virus was detected in >90% of sequenced infections in the area [1,17,18]. B.1.351 infection was corroborated by the experimental findings in this paper that all plasmas bound to B.1.351 spike and RBD, had reduced binding to DMS library variants with mutations to site 484, and better neutralized B.1.351 spike-pseudotyped lentiviral particles relative to D614G particles. All participant samples had detectable antibody binding and neutralizing titers against B.1.351 SARS-CoV-2 spike.

Early-2020 convalescent plasma samples were previously described [27,28] and collected as part of the prospective longitudinal Hospitalized or Ambulatory Adults with Respiratory Viral Infections (HAARVI) cohort study of individuals with SARS-CoV-2 infection in Seattle, WA, between February and July 2020. Written informed consent was obtained from each participant. The plasma samples from 17 individuals were examined here (8 of 17 females; age range 23 to 76 years, mean 51.6 years, median 56 years). These samples were collected approximately 30 days post-symptom onset (mean 31.6 days, median 29 days, min 15 days, max 61 days). Five cases were hospitalized, 2 were asymptomatic, and the remainder were symptomatic non-hospitalized. The neutralization activity of plasma samples before and after depletion of RBD-binding antibodies in **Fig 2** and RBD binding-escape maps in **S4 Fig** were previously reported [28], but neutralization assays for all 30-days post-symptom onset plasmas in **Figs 5 and S5** were newly performed in this study. The neutralization assays on the 100-day early 2020 samples in **S5 Fig** were previously reported [38]. This work was approved by the University of Washington Institutional Review Board (protocol #STUDY00000959).

## Plasma separation from whole blood

Plasma was separated from EDTA-anticoagulated blood by centrifugation at 500 rcf for 10 min and stored at −80°C. Aliquots of plasma samples were heat-inactivated at 56°C for 30 min

and clarified by centrifugation at 10,000 rcf for 5 min, after which the clear middle layer was used for experiments. Inactivated plasma was stored in single-use aliquots to prevent freeze–thaw cycles.

## Construction of B.1.351 RBD yeast-displayed DMS library

Duplicate single-mutant site-saturation variant libraries were designed in the background of the spike receptor binding domain (RBD) from SARS-CoV-2 B.1.351 (identical to that from Wuhan-Hu-1, Genbank accession number MN908947, residues N331-T531, with the addition of the following amino-acid substitutions: K417N, E484K, N501Y), and produced by Twist Bioscience. The Genbank map of the plasmid encoding the unmutated SARS-CoV-2 B.1.351 RBD in the yeast-display vector is available at https://github.com/jbloomlab/SARS-CoV-2-RBD_B.1.351/blob/main/data/plasmid_maps/3021_pETcon-SARS-CoV-2-RBD_K417N_E484K_N501Y.gb. The site-saturation variant libraries were delivered as double-stranded DNA fragments by Twist Bioscience. The final unmutated DNA sequence delivered is:

tctgcaggctagtggtggaggaggctctggtggaggcggCCgcggaggcggagggtcggctagccatatgAATATCACG AACCTTTGTCCTTTCGGTGAGGTCTTCAATGCTACTAGATTCGCATCCGTGTATGC ATGGAATAGAAAGAGAATTAGTAATTGTGTAGCGGACTACTCTGTACTTTATAA CTCCGCCTCCTTCTCCACATTCAAGTGTTACGGTGTATCTCCCACCAAGTTGAATG ATCTATGCTTTACAAACGTTTACGCCGATAGTTTCGTAATTAGAGGCGATGAAG TGCGTCAGATCGCACCAGGCCAGACGGGCAACATAGCAGACTATAATTATA AGCTGCCTGATGACTTCACCGGCTGTGTGATAGCTTGGAACTCAAATAATCTAGA TTCCAAGGTGGGAGGCAATTACAATTATTTGTACCGTCTGTTCCGTAAAAGCAA TTTGAAACCATTTGAAAGAGACATTAGCACTGAAATTTATCAAGCAGGGTCCAC CCCGTGCAACGGCGTAAAGGGCTTTAACTGTTATTTCCCATTACAGTCTTATG GTTTCCAACCTACGTACGGAGTCGGGTATCAGCCGTACAGGGTTGTGGTTCTTTC ATTTGAACTGCTGCACGCGCCCGCAACCGTATGCGGGCCGAAGAAATCAACGctcga ggggggcggttccgaacaaaagcttatttctgaagaggacttgtaatagagatctgataacaacagtgtagatgtaacaaatcgactt tgttcccactgtacttttagctcgtacaaaatacaatatacttttcatttctccgtaaacaacatgtttttcccatgtaatatccttttctatt tttcgttccgttaccaactttacacatactttatatagctattcacttctatacactaaaaaaactaagacaattttaattttgctgcctgccatat ttcaatttgttataaattcctataatttatcctattagtagctaaaaaaagatgaatgtgaatcgaatcctaagagaatt

This sequence has 5' and 3' flanking sequences that are unmutated in the variant libraries (lower case). The uppercase portion is the RBD coding sequence, amino acids N331–T531 (Wuhan-Hu-1 spike numbering). The libraries were designed to contain all 19 amino acids at each site in the RBD, without stop codons, with no more than one amino-acid mutation per variant. The variant gene fragments were PCR-amplified with these primers: 5'-tctgcaggc-tagtggtggag-3' and 5'-agatcggaagagcgtcgtgtagggaaagagtgtagatctcggtggtcgccgtatcattaattctcttag-gattcgattcacattc-3'. (primer-binding regions underlined in the sequence above). A second round of PCR was performed using the same forward primer (5'-tctgcaggctagtggtggag-3') and the reverse primer 5'-ccagtgaattgtaatacgactcactatagggcgaattg-gagctcgcggccgcnnnnnnnnnnnnnnnnnnagatcggaagagcgtcgtgtag-3' to append the Nx16 barcodes and add the overlapping sequences to clone into the recipient vector backbone as described in [25,66].

Failed positions in the Twist-delivered library (sites 362, 501, and 524 in Wuhan-Hu-1 numbering) were mutagenized in-house using a PCR-based method with NNS degenerate primers and cloned into the unmutated wildtype backbone plasmid using NEB HiFi assembly, exactly as described in [66]. These were then PCR-amplified using the same 5'-tctgcaggctagt ggtggag-3' and 5'-ccagtgaattgtaatacgactcactatagggcgaattggagctcgcggccgcnnnnnnnnnnnnnnnn nnnagatcggaagagcgtcgtgtag-3' primers to pool with the barcoded Twist library gene fragments.

The barcoded variant gene fragments were cloned in bulk into the NotI/SacI-digested unmutated wildtype plasmid, as described in [25,66]. The Genbank plasmid map for the fully assembled, barcoded B.1.351 RBD libraries (with the unmutated B.1.351 RBD sequence) is available at https://github.com/jbloomlab/SARS-CoV-2-RBD_B.1.351/blob/main/data/plasmid_maps/pETcon-SARS-CoV-2-RBD-B1351_lib-assembled.gb. The pooled, barcoded mutant libraries were electroporated into *E. coli* (NEB 10-beta electrocompetent cells, New England BioLabs C3020K) and plated at a target bottleneck of 50,000 variants per duplicate library, corresponding to >10 barcodes per mutant within each library. Colonies from bottle-necked transformation plates were scraped and plasmid purified. Plasmid libraries (10 μg plasmid per replicate library) were transformed into the AWY101 yeast strain [67] according to the protocol of Gietz and Schiestl [68].

## PacBio sequencing to link variant mutations and barcodes

As described by Starr et al. [25], PacBio sequencing was used to generate long sequence reads spanning the Nx16 barcode and RBD coding sequence. PacBio sequencing amplicons were prepared from library plasmid pools via NotI digestion, gel purification, and Ampure XP bead clean-up. Sample-specific barcodes and SMRTbells were ligated using the HiFi Express v2 kit. The multiplexed libraries were sequenced on a PacBio Sequel II with a 15-hour movie collection time. Demultiplexed PacBio HiFi circular consensus sequences (CCSs) were generated using the SMRT Link GUI, version 10.1.0.119588. HiFi reads are CCSs with $> = 3$ full passes and a mean quality score $Q > = 20$. The resulting CCSs are available on the NCBI Sequence Read Archive, BioProject PRJNA770094, BioSample SAMN22208699.

HiFi reads were processed using alignparse (version 0.2.6) [69] to determine each variant's mutations and the associated Nx16 barcode sequence, requiring no more than 45 nucleotide mutations from the intended target sequence, an expected 16-nt length barcode sequence, and no more than 4 mismatches across the sequenced portions of the vector backbone. Attribution of barcodes to library variants determined that the libraries contained 3,807 of the 3,819 possible single amino-acid mutations to the B.1.351 RBD. Approximately 26% of barcodes in the duplicate libraries corresponded to wildtype B.1.351 RBD (**S2A Fig**). The libraries were designed to contain only wildtype and 1-amino acid mutations, but some multiple mutations and stop codons were stochastically introduced during the library generation process. These mutations were excluded from downstream analysis of the effects of mutations on ACE2 binding, RBD expression, and plasma antibody binding, except when used in quality control checks (i.e., that most variants containing premature stop codons should not be expressed on the yeast cell surface and thus should have very low expression scores).

## Determining the effects of mutations on RBD expression and ACE2 binding to filter the library for functional variants

The effects of each mutation on RBD expression on the surface of yeast and on ACE2 binding were measured essentially as described previously for the Wuhan-Hu-1 RBD [25]. Specifically, each biological replicate library was grown overnight at 30°C in 45mL SD-CAA media (6.7g/L Yeast Nitrogen Base, 5.0g/L Casamino acids, 1.065 g/L MES acid, and 2% w/v dextrose) at an initial OD600 of 0.4. To induce RBD surface expression, yeast were back-diluted in SG-CAA +0.1%D (2% w/v galactose supplemented with 0.1% dextrose) induction media at 0.67 OD600 and incubated at room temperature for 16–18 hours with mild agitation. For RBD expression experiments, 45 OD units of yeast were labeled in 1:100 diluted chicken-anti-Myc-FITC antibody (Immunology Consultants CMYC45F) to detect the RBD's C-terminal Myc tag. For ACE2-binding experiments, 12 OD units of yeast were incubated overnight at room

temperature with monomeric biotinylated ACE2 (ACROBiosystems AC2-H82E8) across a concentration range of $10^{-13}$ M to $10^{-6}$ M at 1-log intervals. Labeling volumes were increased at low ACE2 concentration to limit ligand depletion effects. Cells were then labeled with 1:100 diluted Myc-FITC to detect RBD expression and 1:200 Streptavidin-PE (Invitrogen S866) to detect binding of biotinylated ACE2.

Cells were processed on a BD FACSAria II and sorted into four bins from low to high RBD expression (measured by myc-FITC staining) or ACE2 binding (measured by streptavidin-PE fluorescence). The RBD expression sort bins were set such that bin 1 would capture 99% of unstained cells, and the remaining 3 bins divide the remainder of each mutant RBD library into equal tertiles. For ACE2 binding, bin 1 captured 95% of cells expressing unmutated RBD incubated with no ACE2 (0 M), and bin 4 captured 95% of cells expressing unmutated RBD incubated with a saturating amount of ACE2 ($10^{-6}$ M). Bins 2 and 3 equally divided the distance between the bin 1 upper and bin 4 lower fluorescence boundaries on a log scale. The frequency of each variant in each bin was determined by Illumina sequencing of RBD variant barcodes.

The effects of each mutation on RBD expression and ACE2 binding were determined as described in [25]. RBD mutant expression and ACE2 binding scores were calculated according to the equations in [25]. For ACE2 binding, a score of –1.0 corresponds to a 10-fold loss in affinity ($K_d$) compared to the wildtype RBD. For RBD expression, a score of –1.0 corresponds to a 10-fold reduction in mean RBD-myc-FITC fluorescence intensity. These measurements were used to computationally filter library variants that were highly deleterious for RBD expression or ACE2 binding and would likely represent spurious antibody-escape mutations (see below for details). The ACE2 binding and RBD expression scores for the single amino-acid mutations in the B.1.351 RBD are available at https://github.com/jbloomlab/SARS-CoV-2-RBD_B.1.351/blob/main/data/final_variant_scores.csv.

As previously described, prior to performing the antibody-escape experiments, the yeast libraries were pre-sorted for RBD expression and binding to dimeric ACE2 (ACROBiosystems AC2-H82E6) to eliminate RBD variants that are completely misfolded or non-functional, such as those lacking modest ACE2 binding affinity [37]. Specifically, unmutated B.1.351 RBD and each RBD mutant library were incubated with dimeric ACE2 at $10^{-8}$ M (a saturating concentration of ACE2 for unmutated B.1.351 RBD). A FACS selection gate was set to capture 98% of cells expressing unmutated B.1.351 RBD that were incubated with $10^{-10}$ M ACE2, to purge the mutant libraries of highly deleterious mutations (i.e., those that have <1% the affinity of unmutated B.1.351 RBD). These pre-sorted yeast libraries containing RBD variants with at least nominal expression and ACE2 binding were used in downstream antibody-escape experiments (see below).

## Depleting plasma of nonspecific yeast-binding antibodies prior to antibody-escape experiments

Prior to the yeast-display deep mutational scanning, plasma samples were twice-depleted of nonspecific yeast-binding antibodies. AWY101 yeast containing a negative control (containing an empty vector pETcon plasmid) were grown overnight at 30˚C in galactose-containing media. Then, up to 50 microliters of plasma samples were incubated, rotating, with 40 OD units of the yeast for 2 hours at room temperature in a total volume of 1mL. The yeast cells were pelleted by centrifugation, and the supernatant was transferred to an additional 40 OD units of yeast cells, and the incubation was repeated overnight at 4˚C. Before beginning the plasma-escape mapping experiments, the negative control yeast were pelleted by centrifugation and the supernatant (containing serum antibodies but not negative control yeast or yeast-binding antibodies) was used in plasma-escape mapping.

## FACS sorting of yeast libraries to select B.1.351 mutants with reduced binding by polyclonal plasmas from B.1.351-convalescent individuals

Plasma mapping experiments were performed in biological duplicate using the independent mutant RBD libraries, similarly to as previously described for monoclonal antibodies [37] and polyclonal plasma samples [28]. Mutant yeast libraries induced to express RBD were washed and incubated with plasma at a range of dilutions for 1 hour at room temperature with gentle agitation. For each plasma, we chose a sub-saturating dilution such that the amount of fluorescent signal due to plasma antibody binding to RBD was approximately equal across samples. The exact dilution used for each plasma is given in **S3 Fig**. After the plasma incubations, the libraries were secondarily labeled for 1 hour with 1:100 fluorescein isothiocyanate-conjugated anti-MYC antibody (Immunology Consultants Lab, CYMC-45F) to label for RBD expression and 1:200 Alexa Fluor-647-conjugated goat anti-human-IgA+IgG+IgM (Jackson ImmunoResearch 109-605-064) to label for bound plasma antibodies. A flow cytometric selection gate was drawn to capture 3–6% of the RBD mutants with the lowest amount of plasma binding for their degree of RBD expression (**S3 Fig**). For each sample, approximately 10 million RBD$^+$ cells (range $10^7$ to $1.5 \times 10^7$ cells) were processed on the BD FACSAria II cell sorter, with between $4 \times 10^5$ and $2 \times 10^6$ plasma-escaped cells collected per sample. Antibody-escaped cells were grown overnight in synthetic defined medium with casamino acids (6.7g/L Yeast Nitrogen Base, 5.0g/L Casamino acids, 1.065 g/L MES acid, and 2% w/v dextrose + 100 U/mL penicillin + 100 μg/mL streptomycin) to expand cells prior to plasmid extraction.

## DNA extraction and Illumina sequencing

Plasmid samples were prepared from 30 optical density (OD) units (1.6e8 colony forming units (cfus)) of pre-selection yeast populations and approximately 5 OD units (~3.2e7 cfus) of overnight cultures of plasma-escaped cells (Zymoprep Yeast Plasmid Miniprep II) as previously described [37]. The 16-nucleotide barcode sequences identifying each RBD variant were amplified by polymerase chain reaction (PCR) and prepared for Illumina sequencing as described in [25]. Specifically, a primer with the sequence 5′-AATGATACGGCGACCACC-GAGA-3′ was used to anneal to the Illumina P5 adaptor sequence, and the PerkinElmer Next-Flex DNA Barcode adaptor primers with the sequence 5′-CAAGCAGAAGACGGCATACGAGATxxxxxxxxGTGACTGGAGTTCA-GACGTGTGCTCTTCCGATCT-3′ (where xxxxxxxx indicates the sample index sequence) were used to anneal to the Illumina P7 adaptor sequence and append sample indexes for sample multiplexing. Barcodes were sequenced on an Illumina HiSeq 2500 with 50 bp single-end reads. To minimize noise from inadequate sequencing coverage, we ensured that each antibody-escape sample had at least 2.5x as many post-filtering sequencing counts as FACS-selected cells, and reference populations had at least 2.5e7 post-filtering sequencing counts.

## Analysis of deep sequencing data to compute each mutation's escape fraction

Escape fractions were computed as described in [37], with minor modifications as noted below. We used the dms_variants package (https://jbloomlab.github.io/dms_variants/, version 0.8.10) to process Illumina sequences into counts of each barcoded RBD variant in each pre-selection and antibody-escape population. For each plasma selection, we computed the escape fraction for each barcoded variant using the deep sequencing counts for each variant in the original and plasma-escape populations and the total fraction of the library that escaped antibody binding via the formula provided in [37]. Specifically:

$E_v = F \times (n_v^{post}/N_{post}) \div (n_v^{pre}/N_{pre})$ where $F$ is the total fraction of the library that escapes antibody binding (these fractions are given as percentages in **S3C Fig**), $n_v^{post}$ and $n_v^{pre}$ are the counts of variant $v$ in the RBD library after and before enriching for antibody-escape variants with a pseudocount of 0.5 added to all counts, and $N_{post} = \sum_v n_v^{post}$ and $N_{pre} = \sum_v n_v^{pre}$ are the total counts of all variants after and before the antibody-escape enrichment.

These escape fractions represent the estimated fraction of cells expressing that specific variant that falls in the escape bin, such that a value of 0 means the variant is always bound by plasma and a value of 1 means that it always escapes plasma binding.

We then applied a computational filter to remove variants with >1 amino-acid mutation, low sequencing counts, or highly deleterious mutations that might cause antibody escape simply by leading to poor expression of properly folded RBD on the yeast cell surface [25,37]. Specifically, we removed variants that had ACE2 binding scores < −3.0 or expression scores < −1.0, after calculating mutation-level deep mutational scanning scores for this library as in [25]. An ACE2 binding score threshold of −3.0 retained 99.4% and an RBD expression score threshold of −1.0 retained 93.8% of all RBD mutations observed > = 50x in GISAID as of Aug. 1, 2021 (**S2C Fig**).

We also removed all mutations where the wildtype residue was a cysteine. There were 2,014 out of the possible 3,653 mutations to non-disulfide bond residues in the RBD that passed these computational filters.

The reported antibody-escape scores throughout the paper are the average across the libraries; these scores are also in **S3 Data**. Correlations in final single-mutant escape scores are shown in **S3D Fig**.

For plotting and analyses that required identifying RBD sites of strong escape, we considered a site to mediate strong escape if the total escape (sum of mutation-level escape fractions) for that site exceeded the median across sites by >5-fold, and was at least 5% of the maximum for any site. Full documentation of the computational analysis is at https://github.com/jbloomlab/SARS-CoV-2-RBD_B.1.351.

## Differences between composition and analysis of B.1.351 RBD libraries and Wuhan-Hu-1 libraries

Importantly, because the B.1.351 libraries were generated using a different method than the Wuhan-Hu-1 RBD libraries, which is fully described in [25], the analysis of deep sequencing data to compute each mutation's escape fraction is also different. The newly generated B.1.351 libraries were ordered from Twist Bioscience to have one amino-acid mutation per variant, whereas the Wuhan-Hu-1 libraries were generated in-house with a PCR-based approach, with an average of 2.7 mutations per variant [25]. Because there were often multiple mutations per variant for the Wuhan-Hu-1 libraries, global epistasis modeling was used to deconvolve the effects of single amino-acid mutations on antibody binding [28,37], whereas for the B.1.351 libraries, the measurements for single-mutant variants were used directly (occasional variants with multiple mutations were discarded) to calculate antibody escape.

## Generation of pseudotyped lentiviral particles

HEK-293T (American Type Culture Collection, CRL-3216) cells were used to generate SARS-CoV-2 spike-pseudotyped lentiviral particles and 293T-ACE2 cells (Biodefense and Emerging Infectious Research Resources Repository (BEI Resources), NR-52511) were used to titer the SARS-CoV-2 spike-pseudotyped lentiviral particles and to perform neutralization assays (see below).

For experiments involving D614G spike, we used spike-pseudotyped lentiviral particles that were generated essentially as described in [70], using a codon-optimized SARS-CoV-2 spike from Wuhan-Hu-1 strain that contains a 21-amino-acid deletion at the end of the cytoplasmic tail [27] and the D614G mutation that is now predominant in human SARS-CoV-2 [30]. The plasmid encoding this spike, HDM_Spikedelta21_D614G, is available from Addgene (#158762) and BEI Resources (NR-53765), and the full sequence is at (https://www.addgene.org/158762). Point mutations were introduced into the RBD of this plasmid via site-directed mutagenesis.

For experiments involving B.1.351 spike, we introduced the following mutations into the HDM_Spikedelta21_D614G plasmid to match the amino acid sequence of EPI_ISL_700420: 80A, D215G, L242-244del, K417N, E484K, N501Y, and A701V. This plasmid map is available online at https://github.com/jbloomlab/SARS-CoV-2-RBD_B.1.351/blob/main/data/plasmid_maps/2957_HDM_Spikedelta21_B.1.351.gb.

To generate spike-pseudotyped lentiviral particles [70], $6 \times 10^5$ HEK-293T (ATCC CRL-3216) cells per well were seeded in 6-well plates in 2 mL D10 growth media (Dulbecco's Modified Eagle Medium with 10% heat-inactivated fetal bovine serum, 2 mM l-glutamine, 100 U/mL penicillin, and 100 μg/mL streptomycin). 24 hours later, cells were transfected using BioT transfection reagent (Bioland Scientific) with a Luciferase_IRES_ZsGreen backbone, Gag/Pol lentiviral helper plasmid (BEI Resources NR-52517), and wild-type or mutant SARS-CoV-2 spike plasmids. Media was changed to fresh D10 at 24 hours post-transfection. At ~60 hours post-transfection, viral supernatants were collected, filtered through a 0.45 μm surfactant-free cellulose acetate low protein-binding filter, and stored at −80˚C.

## Titering of pseudotyped lentiviral particles

Titers of spike-pseudotyped lentiviral particles were determined as described in [70] with the following modifications. 100 μL of diluted spike-pseudotyped lentiviral particles was added to 1.25e4 293T-ACE2 cells (BEI Resources NR-52511), grown overnight in 50 μL of D10 growth media in a 96-well black-walled poly-L-lysine coated plate (Greiner Bio-One, 655936). Relative luciferase units (RLU) were measured 65 hours post-infection (Promega Bright-Glo, E2620) in the infection plates with a black back-sticker (Thermo Fisher Scientific, NC9425162) added to minimize background. Titers were first estimated from the average of 8 two-fold serial dilutions of virus starting at 10 μL virus in a total volume of 150 μL, performed in duplicate.

## Neutralization assays

293T-ACE2 cells (BEI Resources NR-52511) were seeded at 1.25e4 cells per well in 50 μL D10 in poly-L-lysine coated, black-walled, 96-well plates (Greiner 655930). 24 hours later, pseudotyped lentivirus supernatants were diluted to ~200,000 RLU per well (determined by titering as described above) and incubated with a range of dilutions of plasma for 1 hour at 37˚C. 100 μL of the virus-antibody mixture was then added to cells. At about 50 or ∼70 hours post-infection, luciferase activity was measured using the Bright-Glo Luciferase Assay System (Promega, E2610). Fraction infectivity of each plasma antibody-containing well was calculated relative to a no-plasma well inoculated with the same initial viral supernatant in the same row of the plate. We used the neutcurve package (https://jbloomlab.github.io/neutcurve version 0.5.7) to calculate the inhibitory concentration 50% ($IC_{50}$) and the neutralization titer 50% ($NT_{50}$), which is $1/IC_{50}$, of each plasma against each virus by fitting a Hill curve with the bottom fixed at 0 and the top fixed at 1.

## Depletion of RBD-binding antibodies from polyclonal sera

Two rounds of sequential depletion of RBD-binding antibodies were performed for vaccine-elicited sera. Magnetic beads conjugated to the SARS-CoV-2 B.1.351 RBD (ACROBiosystems, MBS-K032) were prepared according to the manufacturer's protocol. Beads were resuspended in ultrapure water at 1 mg beads/mL and a magnet was used to wash the beads 3 times in phosphate-buffered saline (PBS) with 0.05% bovine serum albumin (BSA). Beads were then resuspended in PBS with 0.05% BSA at 1 mg beads per mL. Beads (manufacturer-reported binding capacity of 10–40 µg/mL anti-RBD antibodies) were incubated with human plasma at a 2:1 ratio beads:plasma, rotating overnight at 4˚C or for 2 hours at room temperature. A magnet (MagnaRack Magnetic Separation Rack, Thermo Fisher Scientific, CS15000) was used to separate antibodies that bind RBD from the supernatant, and the supernatant (the post-RBD antibody depletion sample) was removed. A mock depletion (pre-depletion sample) was performed by adding an equivalent volume of PBS + 0.05% BSA and rotating overnight at 4˚C or for 2 hours at room temperature. Up to three rounds of depletions were performed to ensure full depletion of RBD-binding antibodies. For the neutralization assays on these plasmas depleted of RBD-binding antibodies, the reported plasma dilution is corrected for the dilution incurred by the depletion process. Note that these assays were performed in 293T cells over-expressing human ACE2, which may underestimate contributions of non-RBD-binding antibodies to viral neutralization [7,35,60].

## Measurement of plasma binding to RBD or spike by enzyme-linked immunosorbent assay (ELISA)

The IgG ELISAs for spike protein and RBD were conducted as previously described [71]. Briefly, ELISA plates were coated with recombinant B.1.351 spike (purified and prepared as described in [71]) and RBD (ACROBiosystems, SPD-C52Hp) antigens described in at 2 µg/mL. Five 3-fold serial dilutions of sera beginning at 1:500 were performed in PBS with 0.1% Tween with 1% Carnation nonfat dry milk. Dilution series of the synthetic sera comprised of the anti-RBD antibody REGN10987 [72], which binds to both Wuhan-1-like RBD and B.1.351 RBD, and pooled pre-pandemic human serum from 2017–2018 (Gemini Biosciences; nos. 100–110, lot H86W03J; pooled from 75 donors) were performed such that the anti-spike antibody was present at a highest concentration of 0.25 µg/mL. REGN10987 was recombinantly produced by Genscript. The REGN10987 is the same as that used in [73]. Pre-pandemic serum alone, without anti-RBD antibody depletion, was used as a negative control, averaged over 2 replicates. Secondary labeling was performed with goat anti-human IgG-Fc horseradish peroxidase (HRP) (1:3000, Bethyl Labs, A80-104P). Antibody binding was detected with TMB/E HRP substrate (Millipore Sigma, ES001) and 1 N HCl was used to stop the reaction. $OD_{450}$ was read on a Tecan infinite M1000Pro plate reader.

## Data visualization

The static logo plot visualizations of the escape maps in the paper figures were created using the dmslogo package (https://jbloomlab.github.io/dmslogo, version 0.6.2) and in all cases the height of each letter indicates the escape fraction for that amino-acid mutation calculated as described above. For each sample, the y-axis is scaled to be the greatest of (a) the maximum site-wise escape metric observed for that sample, (b) 20x the median site-wise escape fraction observed across all sites for that plasma, or (c) an absolute value of 1.0 (to appropriately scale samples that are not noisy but for which no mutation has a strong effect on antibody binding). Sites K417, L452, S477, T478, E484, and N501 have been added to logo plots due to their

frequencies among circulating viruses. The code that generates these logo plot visualizations is available at https://github.com/jbloomlab/SARS-CoV-2-RBD_B.1.351/blob/main/results/summary/escape_profiles.md. In many of the visualizations, the RBD sites are categorized by epitope region [23] and colored accordingly. We define the class 1 epitope as residues 403+405 +406+417+420+421+453+455–460+473–476+486+487+489+504, the class 2 epitope as residues 472+483–485+490–494, the class 3 epitope to be residues 345+346+437–452+496+498–501, and the class 4 epitope as residues 365–372+378+382–386.

For the static structural visualizations in the paper figures, the RBD surface (PDB 6M0J) was colored by the site-wise escape metric at each site, with white indicating no escape and red scaled to be the same maximum used to scale the y-axis in the logo plot escape maps, determined as described above. We created interactive structure-based visualizations of the escape maps using dms-view [74] that are available at https://jbloomlab.github.io/SARS-CoV-2-RBD_B.1.351/. The logo plots in these escape maps can be colored according to the deep mutational scanning measurements of how mutations affect ACE2 binding or RBD expression as described above.

## Statistical analysis

The percent of neutralizing activity of early-2020 and B.1.351-convalescent plasmas due to RBD-binding antibodies is plotted with the plotnine python package, version 0.8.0 (https://plotnine.readthedocs.io/en/stable/index.html), shown as a Tukey boxplot (middle line indicating median, box limits indicating interquartile range) with individual measurements overlaid as points. P-values are from a log-rank test accounting for censoring, calculated with the lifelines python package, version 0.25.10 (https://lifelines.readthedocs.io/en/latest/).

## Supporting information

**S1 Fig. Enzyme-linked immunosorbent assay (ELISA) and neutralization curves of B.1.351 convalescent plasmas before and after depletion of B.1.351 RBD-binding antibodies. (A)** Controls showing that the RBD antibody depletion completely removes a RBD-targeting neutralizing antibody. Effect of two rounds of RBD antibody depletion on binding to B.1.351 RBD and spike (left) and neutralization of B.1.351 spike-pseudotyped lentiviral particles (right) by synthetic serum. The synthetic serum was made by adding the RBD-targeting antibody REGN10987 that binds both Wuhan-Hu-1 and B.1.351 RBD [72] to pre-pandemic pooled serum at 50 μg/mL. The x-axis indicates the antibody concentration (μg/mL), and the y-axis is the optical density at wavelength 450 (OD450) reading at each dilution (left) or fraction infectivity (right). **(B)** Binding of B.1.351 convalescent plasmas to B.1.351 RBD and spike for mock depletion (gray lines) and depletion of RBD-binding antibodies (orange lines). Some samples were depleted three times (dashed lines and open circles) if two rounds of depletion did not abrogate binding to RBD. There were not substantial reductions in OD450 after the third round of depletions, so we reasoned that the samples were maximally depleted, and no further rounds of depletions were performed. Removal of RBD-binding antibodies only modestly reduces spike binding, consistent with prior findings that the majority of anti-spike antibodies do not bind the RBD [28,77–80]. **(C)** Neutralization curves for plasma mock depletion (gray circles) and depletion of RBD-binding antibodies (orange triangles). Each assay was performed in technical duplicate, and points show the mean and standard error of the replicates. Pre-pandemic pooled serum was included in (B) and (C) as a negative control for binding and neutralization. RBD-binding antibodies were removed from the plasma using streptavidin magnetic beads conjugated to biotinylated B.1.351 RBD. All binding assays were performed with B.1.351 RBD and spike, and all neutralization assays were performed with B.1.351 spike-pseudotyped

lentiviral particles. This figure shows the underlying measurements for all of the B.1.351 plasmas in **Fig 2**; the underlying measurements for the early 2020 plasmas in **Fig 2** are shown in [28].
(EPS)

**S2 Fig. Generation of the B.1.351 RBD mutant libraries and measurements of effects of mutations on ACE2 binding and RBD expression. (A)** Schematic showing the B.1.351 RBD mutant library design. A site-saturation variant library was generated in the B.1.351 RBD background, targeting one amino-acid mutation per variant. Nx16 unique DNA barcodes were added to the variant gene fragments. The Nx16 barcodes were linked to their associated RBD mutations by PacBio circular consensus sequencing (CCS). The plasmid library DNA was transformed into yeast cells. In downstream experiments, the Nx16 barcodes are sequenced by short-read Illumina sequencing. The tables at right indicate key library statistics. **(B)** Correlations between biological independent replicate library measurements of the effects of single mutations on ACE2 binding and RBD expression, measured as described in [25]. See **Methods** for experimental details. **(C)** Thresholds on the ACE2 binding and RBD expression scores (dashed orange lines) for the B.1.351 mutant library to computationally filter highly deleterious variants that may represent spurious antibody-escape mutations. Importantly, we aimed to retain most mutations that have been observed $> = 50$ times in sequenced SARS-CoV-2 isolates. The x-axis categorizes mutations by their number of observations in GISAID [81] as of Aug. 1, 2021. An ACE2 binding score threshold of $> = -3.0$ (1,000-fold loss in binding affinity) and an RBD expression score of $> = -1.0$ (10-fold loss in RBD expression) were chosen, which filter comparable numbers of mutations as in prior Wuhan-Hu-1 experiments [28,73]. These filters retain 99.4 and 93.8% of mutations, respectively, that have been observed $> = 50$ times in sequenced SARS-CoV-2 isolates. **(D)** Relationship between the ACE2 binding and RBD expression scores for the B.1.351 RBD library compared to those previously published for the Wuhan-Hu-1 library [25]. The computational filters used for antibody-escape experiments for the Wuhan-Hu-1 [28] and B.1.351 libraries are dashed orange lines. Each dot is one mutation, and mutations to disulfide bonds are shown in red. A key difference is that for the previously published Wuhan-Hu-1 experiments, dimeric rather than monomeric ACE2 was used [25].
(EPS)

**S3 Fig. Deep mutational scanning approach to map mutations that reduce binding of B.1.351 infection-elicited polyclonal plasma antibodies to the B.1.351 RBD. (A)** Schematic of the approach. The RBD is expressed on the surface of yeast (top left). Flow cytometry is used to quantify both RBD expression (via a C-terminal MYC tag, green star) and antibody binding to the RBD protein expressed on the surface of each yeast cell (bottom left). A library of yeast expressing B.1.351 RBD mutants was incubated with convalescent plasmas and fluorescence-activated cell sorting (FACS) was used to enrich for cells expressing RBD that bound reduced amounts of plasma antibodies, as detected using an IgA+IgG+IgM secondary antibody. Deep sequencing was used to quantify the frequency of each mutation in the initial and antibody-escape cell populations. We quantified the effect of each mutation as the escape fraction, which represents the fraction of cells expressing RBD with that mutation that fell in the antibody escape FACS bin. Escape fractions are represented in logo plots, with the height of each letter proportional to the effect of that amino-acid mutation on antibody binding. The site-level escape metric is the sum of the escape fractions of all mutations at a site. Experimental and computational filtering were used to remove RBD mutants that were misfolded or unable to bind the ACE2 receptor. **(B)** Left: Representative plots of nested FACS gating strategy used for all plasma selection experiments to select for single cells. Samples were gated by SSC-A versus FSC-A, SSC-W versus SSC-H, and FSC-W versus FSC-H) that also express RBD

(FITC-A vs. FSC-A). Right: The RBD mutant libraries were sorted to retain cells expressing variants that bound to ACE2 with at least nominal affinity. Unmutated B.1.351 RBD and each RBD mutant library was incubated with dimeric ACE2 at $10^{-8}$ M. A FACS selection gate was set to capture 98% of cells expressing unmutated B.1.351 RBD that were incubated with $10^{-10}$ M ACE2, to purge the mutant libraries of highly deleterious mutations (i.e., those that have <1% the affinity of unmutated B.1.351 RBD). **(C)** Left: FACS gating strategy for one of two independent libraries to select cells expressing RBD mutants with reduced binding by poly-clonal sera (cells in blue). Gates were set manually during sorting. Selection gates were set to capture ~5% of the RBD+ library. The same gate was set for both independent libraries stained with each plasma sample, and the FACS scatter plots looked qualitatively similar between the two libraries. Right: the fraction of library cells that fall into each selection gate. **(D)** Mutation- and site-level correlations of escape scores between biologically independent library replicates. SSC-A, side scatter-area; FSC-A, forward scatter-area; SSC-W, side scatter-width; SSC-H, side scatter-height; FSC-W, forward scatter-width; FSC-H, forward scatter height; FITC-A, fluores-cein isothiocyanate-area.
(EPS)

**S4 Fig. Escape maps for the early 2020 convalescent plasmas, as measured using a deep mutational scanning approach in the Wuhan-Hu-1 RBD background.** The line plots at left indicate the site-level antibody escape for all RBD sites, and the logo plots at right zoom in on key sites (highlighted in purple on the line plot x-axes). For each sample, the y-axis is scaled independently. RBD sites are colored by antibody epitope. Sites 417, 484, and 501 are labeled with red text on the x-axis. All 11 samples from the Washington State early 2020 cohort [28] are shown here and averaged in **Fig 4**. Interactive versions of logo plots and structural visuali-zations are at https://jbloomlab.github.io/SARS-CoV-2-RBD_B.1.351/. These data were origi-nally published in [28] and are reanalyzed here. The numerical antibody-escape scores are in **S4 Data** and at https://github.com/jbloomlab/SARS-CoV-2-RBD_B.1.351/blob/main/results/prior_DMS_data/early2020_escape_fracs.csv.
(EPS)

**S5 Fig. Neutralization of point mutants of B.1.351 and D614G spike-pseudotyped lenti-viral particles by convalescent plasmas from B.1.351 and early 2020-infected individuals.** **(A)** Each plot shows the neutralization curves of one point mutant and the wildtype measured on the same assay date for each plasma (the same wildtype curve is repeated on multiple plots for comparison). Plots are grouped by assay date. Each point is the average of two technical replicates. **(B)** The fold-decrease in neutralization for samples shown in **Fig 5**, with the addi-tion of previously measured neutralization by samples from 6 early 2020 convalescent individ-uals collected approximately 100 days post-symptom onset [38]. **(C)** The fold-change in IC50 (left) or the absolute IC50 (right) for the neutralization of each point mutant by each plasma. The fold-change IC50 is calculated relative to the geometric mean of two wildtype technical replicates performed on the same assay date. Each point is one technical replicate. The dashed gray line indicates the geometric mean of all wildtype measurements for that plasma, and the orange line indicates the geometric mean of the effect of removing all RBD-binding antibodies. B.1.351 plasma names are prefixed with K*, and early 2020 plasmas are prefixed with "partici-pant". All assays were performed with the "homologous" virus: B.1.351 spike for B.1.351 plas-mas, and D614G spike for early 2020 plasmas. Mutations are given the same names for B.1.351 and D614G spikes, so 417K/N is 417N in the B.1.351 background and 417K in the D614G background; 484E/K is 484E in B.1.351 and 484K in D614G; 501N/Y is 501N in B.1.351 and 501Y in D614G; and 417-484-501 is 417K-484E-501N in B.1.351 and 417N-484K-501Y in

D614G.
(EPS)

**S1 Data. Neutralization titers for B.1.351 and early 2020 infection-elicited sera before and after depletion of homologous RBD-binding antibodies.** This file is also available at: https://github.com/jbloomlab/SARS-CoV-2-RBD_B.1.351/blob/main/experimental_data/results/rbd_depletion_neuts/RBD_depletion_NT50_b1351_haarvi.csv
(CSV)

**S2 Data. The effects of all single amino-acid mutations in the B.1.351 RBD on ACE2 binding and RBD expression.** This file is also available at: https://github.com/jbloomlab/SARS-CoV-2-RBD_B.1.351/blob/main/data/final_variant_scores.csv
(CSV)

**S3 Data. Plasma-escape scores for B.1.351 plasmas against the B.1.351 RBD deep mutational scanning library.** This file is also available at: https://github.com/jbloomlab/SARS-CoV-2-RBD_B.1.351/blob/main/results/supp_data/B1351_raw_data.csv
(CSV)

**S4 Data. Plasma-escape scores for early 2020 plasmas against the Wuhan-Hu-1 RBD deep mutational scanning library.** This file is also available at: https://github.com/jbloomlab/SARS-CoV-2-RBD_B.1.351/blob/main/results/prior_DMS_data/early2020_escape_fracs.csv
(CSV)

**S5 Data. Neutralization titers of early 2020 and B.1.351 plasmas against spike-pseudotyped lentiviral particles in the homologous spike background.** This file is also available at: https://github.com/jbloomlab/SARS-CoV-2-RBD_B.1.351/blob/main/experimental_data/results/neut_titers/neut_titers.csv.
(CSV)

## Acknowledgments

We thank Cathy Lin for administrative support; Dolores Covarrubias, Andy Marty, the Genomics and Flow Cytometry core facilities at the Fred Hutchinson Cancer Research Center, and Katy Munson at the University of Washington PacBio Sequencing Services for experimental support. We also thank all study participants for their generous participation and contribution to this work.

The content is solely the responsibility of the authors and does not necessarily represent the official views of the US government or the other sponsors.

## Author Contributions

**Conceptualization:** Allison J. Greaney, Alex Sigal, Jesse D. Bloom.

**Formal analysis:** Allison J. Greaney.

**Funding acquisition:** Jesse D. Bloom.

**Investigation:** Allison J. Greaney, Rachel T. Eguia.

**Methodology:** Allison J. Greaney, Tyler N. Starr, Andrea N. Loes.

**Resources:** Khadija Khan, Farina Karim, Sandile Cele, John E. Bowen, Jennifer K. Logue, Davide Corti, David Veesler, Helen Y. Chu, Alex Sigal.

**Software:** Allison J. Greaney, Jesse D. Bloom.

**Supervision:** Alex Sigal, Jesse D. Bloom.

**Validation:** Allison J. Greaney, Rachel T. Eguia, Andrea N. Loes.

**Visualization:** Allison J. Greaney.

**Writing – original draft:** Allison J. Greaney, Jesse D. Bloom.

**Writing – review & editing:** Tyler N. Starr, Rachel T. Eguia, Andrea N. Loes, Khadija Khan, Farina Karim, Sandile Cele, John E. Bowen, Jennifer K. Logue, Davide Corti, David Veesler, Helen Y. Chu, Alex Sigal, Jesse D. Bloom.

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
