## [Decision Letter · Decision Letter 0]

2 Dec 2021

Dear Dr Bloom,

Thank you very much for submitting your manuscript "A SARS-CoV-2 variant elicits an antibody response with a shifted immunodominance hierarchy" for consideration at PLOS Pathogens. As with all papers reviewed by the journal, your manuscript was reviewed by members of the editorial board and by several independent reviewers. The reviewers appreciated the attention to an important topic. Based on the reviews, we are likely to accept this manuscript for publication, providing that you modify the manuscript according to the review recommendations.

You can see that the reviewers were very supportive and had only minor comments/requests.

Sincerely,

Sabra L. Klein

Associate Editor

PLOS Pathogens

Andrew Pekosz

Section Editor

PLOS Pathogens

Kasturi Haldar

Editor-in-Chief

PLOS Pathogens

orcid.org/0000-0001-5065-158X

Michael Malim

Editor-in-Chief

PLOS Pathogens

orcid.org/0000-0002-7699-2064

Reviewer Comments (if any, and for reference):

Reviewer's Responses to Questions

**Part I - Summary**

Reviewer #1: Greaney et al. describe experiments comparing the binding and neutralizing antibody specificities in convalescent plasma from 9 persons infected with B.1.351 SARS-CoV-2 variants in the Republic of South Africa to convalescent plasma from 17 persons infected with early 2020 SARS-CoV-2 variants in the US. In lentiviral pseudovirus neutralization assays using ACE2-293T target cells, the authors show that depleting plasma of anti-RBD antibodies removes most of the neutralization activity in the B.1.351 convalescent plasma, as was seen previously for early 2020 variant convalescent plasma. Using the deep mutational scanning method to evaluate the effects of most possible single mutations in the RBD on antibody binding in B.1.351 plasma, they further show that 4/9 samples were predominantly affected by 484 mutation but 3/9 samples were also significantly impacted by 443-450, 498-501, as well as 484 mutations. This contrasted findings from 11 early 2020 plasma samples that were predominantly impacted by 484, 486, and 456 mutations. Neutralization experiments also showed that early 2020 convalescent plasma was greatly affected by E484K/Q and K417N-E484K-N501Y mutations while the impacts of these mutations were more modest for B.1.351 convalescent plasma. B.1.351 plasma samples also tended to be more affected by the G446V mutation. The authors conclude that SARS-CoV-2 variants can elicit polyclonal antibodies with different immunodominance hierarchies.

Experimental methods are strong, innovative, and carefully documented, with no major weaknesses. Conclusions are supported by the data. The authors acknowledge the major limitations of their studies, including use of a neutralization assay that detect mostly RBD-directed antibodies, RBD binding assays that do not completely mimic the trimeric Spike on virions, and the relatively small number of samples. The findings offer a valuable extension of prior studies by providing more detailed characterization of differences in antibody specificities elicited by B.1.351 SARS-CoV-2 infections compared to early 2020 SARS-CoV-2 variant infections. These results have implications for our understanding of immunity elicited by different SARS-CoV-2 variants.

Reviewer #2: Greaney et al utilized a previously established, high throughput and unbiased mutagenesis system to study plasma antibody interactions with neutralizing SARS-CoV-2 viral epitopes located on the receptor binding domain of the Spike protein. They have previously applied this method with great success to an early strain of SARS-CoV-2. Here, they apply the same system to convalescent plasma from individuals infected with the beta variant, B.1.351, and identify differential antibody binding preferences and neutralizing specificities. While this reviewer is enthusiastic about the presentation and analysis of their data sets, some of the caveats of the study are not sufficiently emphasized. Overall, however, the study provides important data on serum antibody recognition of the beta variant.

**Part II – Major Issues: Key Experiments Required for Acceptance**

Reviewer #1: (No Response)

Reviewer #2: none

**Part III – Minor Issues: Editorial and Data Presentation Modifications**

Reviewer #1: (No Response)

Reviewer #2: Please comment in the discussion about how the arbitrary threshold of 3-6% gating might bias the results, particularly in the context of comparison of their new data sets with their old data sets. I note there is a mix of linear and curved gates, with the linear gating strongly favoring variants with low RBD expression. Some of the curved gates do as well.

Please briefly mention why 484 -> V, F, P, L etc weren't included in the pseudotyped virus neutralization analysis, since those mutations appeared to cause the most consistent reduction in antibody binding to RBD based on the mutational scanning. Correspondingly, the authors mention “but within the RBD, site 484 is less immunodominant for B.1.351-elicited plasmas” - presumably this refers only to the effect on neutralization. If so, this claim may need to be revised or substantiated by measurements of neutralization against pseudotyped viruses expressing V, F, P, or L mutations.

Why is L452R excluded from fig 5? The authors show the NT data in the supplement with all of the others, but it isn't in the main body.

No tested mutation, nor the 417-484-501 triple mutant, reduced neutralization by the B.1.351 plasmas as much as removing all RBD-binding antibodies (Fig. 5) – could the authors expand on the significance of this finding in the discussion? Does this have to do with elevated RBD polyclonality in the B.1.351 plasmas? Would this be a virus effect or could it be host response effect?

The claim that there are significant differences in the RBD epitope hierarchies between the B.1.351 and 2020 sera should be tempered or made more specific. There are a couple of reasons for this. First, comparing the escape maps reveals that some responses are actually more similar than different, eg K007/K031/K040 vs participant C and K046/K114/K119 vs participants G and H. There are thus heterogenous subgroups, so definitive differences would need a larger cohort study to overcome potential sampling artifacts. Another extremely important caveat, which is not sufficiently emphasized, are the likely differences in host factors. These factors include ancestry (and thus HLA genetics, which certainly can influence epitope selection), immune response history (including due to differences in microbiota), and prior exposures to endemic coronaviruses (and thus original antigenic sin).

Figure 3 is not called out in the main text.

Minor suggestions:

• Would be nice to have 1-2 sentences in the main text about how the escape system works (yeast display, etc) and how the library was generated.

• Comment on the advantages/challenges of combinatorial sublibraries (all variants in the context of an 484 mutant for example).

• How were 11 of the 17 2020 samples selected for escape mapping?

• Note in main text that escape mapping involved combined detection of IgG/A/M.

• Please provide the formula cited in reference 37 (so readers don’t have to look it up).

• Can the same epitope class coloring be used below the escape fraction plots? Currently just pink and too small.

• Text in Fig S5 is illegible.

PLOS authors have the option to publish the peer review history of their article (what does this mean?). If published, this will include your full peer review and any attached files.

Reviewer #1: No

Reviewer #2: No

Figure Files:

Data Requirements:

Reproducibility:

References:

---

## [Editor Report · Decision Letter 1]

6 Jan 2022

Dear Dr Bloom,

We are pleased to inform you that your manuscript 'A SARS-CoV-2 variant elicits an antibody response with a shifted immunodominance hierarchy' has been provisionally accepted for publication in PLOS Pathogens.

Best regards,

Sabra L. Klein

Associate Editor

PLOS Pathogens

Andrew Pekosz

Section Editor

PLOS Pathogens

Kasturi Haldar

Editor-in-Chief

PLOS Pathogens

orcid.org/0000-0001-5065-158X

Michael Malim

Editor-in-Chief

PLOS Pathogens

orcid.org/0000-0002-7699-2064

Thank you for being so responsive to the reviews. This is excellent work.
---

## [Editor Report · Acceptance letter]

1 Feb 2022

Dear Dr Bloom,

We are delighted to inform you that your manuscript, "A SARS-CoV-2 variant elicits an antibody response with a shifted immunodominance hierarchy," has been formally accepted for publication in PLOS Pathogens.

Best regards,

Kasturi Haldar

Editor-in-Chief

PLOS Pathogens

orcid.org/0000-0001-5065-158X

Michael Malim

Editor-in-Chief

PLOS Pathogens

orcid.org/0000-0002-7699-2064